# Unconscious Integration of Categorical Relationship of Two Subliminal Numbers in Comparison with “5”

**DOI:** 10.3390/bs14040296

**Published:** 2024-04-04

**Authors:** Changjun Li, Qingying Liu, Yingjuan Liu, Jerwen Jou, Shen Tu

**Affiliations:** 1Applied Psychology, School of Politics and Law, Hebei North University, Zhangjiakou 075000, China; 2Applied Psychology, School of Public Administration, Guizhou University of Finance and Economics, Guiyang 550025, China; 3Department of Psychological Science, University of Texas, Rio Grande Valley, Edinburg, TX 78539, USA; 4Institute of Security Development and Modernized Governance, Guizhou University of Finance and Economics, Guiyang 550025, China

**Keywords:** number, categorization, unconscious integration

## Abstract

Many studies have shown that the brain can process subliminal numerals, i.e., participants can categorize a subliminal number into two categories: greater than 5 or less than 5. In the context of many studies on the unconscious integration of multiple subliminal stimuli, the issue of whether multiple subliminal numbers can be integrated is contentious. The same-different task is regarded as a perfect tool to explore unconscious integration. In the two experiments reported, we used a same-different task in which a pair of masked prime numbers was followed by a pair of target numbers, and participants were asked to decide whether the two target numbers were on the same (both smaller or larger than 5) or different sides (one smaller, the other larger than 5) of 5 in magnitude. The results indicated that the prime numbers could be categorized unconsciously, which was reflected by the category priming effect, and that the unconscious category relationship of the two prime numbers could affect the judgment on the category relationship of the two target numbers, as reflected by the response priming effect. The duration of the prime-to-target interstimulus interval (ISI) was also manipulated, showing a positive compatibility effect (PCE) of category priming and a negative compatibility effect (NCE) of response priming no matter whether the ISI was short (50 ms) or long (150 ms). The NCE, which occurred when the prime-to-target ISI was relatively short in this study, contradicted the conventional view but was consistent with previous results of unconscious integration based on an attention modulation mechanism. Importantly, this study provided evidence for the still-under-debate issue of numerical information integration.

## 1. Introduction

The scope of unconscious processing has been studied for over a century [1]. However, there has been no consensus for a long time on the issue of the level and scope of information that can be unconsciously processed. Some researchers believe that the scope of unconscious information processing is very limited in that only simple stimuli can be processed unconsciously [2]. Others argue that any level of conscious processing can be completed unconsciously [3]. However, the hypothesis that the processing range of unconscious information is equivalent to that of conscious processing has yet to be verified. With the advancing of the study of unconscious information processing, the assumption that only simple stimuli can be processed unconsciously has been seriously questioned recently. For example, there is evidence that unconscious information integration can occur involving the relationship between multiple unconscious components [4,5]. One common type of semantic information frequently used in daily life is numbers. This study investigated the question of whether numerical information could be unconsciously integrated.

### 1.1. Unconscious Processing of Numbers

A lot of studies have been devoted to exploring how we process numerals [6,7,8,9,10]. Combined with the methods of studying unconscious processing, a large number of studies have found that the brain can process the semantics of the subliminal numerals. In a series of unconscious priming experiments conducted by Dehaene et al., participants were asked to judge whether the target number was larger or smaller than 5. No matter how the format of the number (in Arabic or spelled-out form) in the prime and target varied, participants responded significantly faster when the prime was congruent with the target (both larger than 5 or smaller than 5) compared with the incongruent condition [11]. Naccache and Dehaene [12] further verified the unconscious access to the semantics of the masked numbers in a similar unconscious priming experiment after excluding the possibility of a stimulus–response connection.

The above research was limited to the unconscious extraction of single-digit semantics. However, some researchers have also found that people can unconsciously integrate the meanings of two subliminal numbers. Unconscious integration of information is defined as a process that generates a new representation from two or more unconscious component representations [4,5], which involves processing the relationship between multiple unconscious components. Doing arithmetic (e.g., addition and multiplication) is a typical example of a numerical integration in which a new representation (the sum or product of the result of an addition or multiplication) is generated from two or more component representations (e.g., addends or multiplicands). There is evidence that when the unconscious prime was a simple addition or subtraction [8,13] or multiplication [9], participants responded faster when the target number matched the result of the addition or multiplication than when the target number did not. In other words, the response to the target was affected by the integration between two unconscious prime numbers. This brings the understanding of the unconscious processing of numerals to a deeper level. Nonetheless, the view of unconscious arithmetic was questioned recently. Some researchers argue that the current evidence for unconscious arithmetic is inconclusive [14]. This doubt has called the theory of unconscious integration into question.

Overall, it seems that the extraction of digital semantics itself can be performed unconsciously but the evidence of unconscious arithmetic reflecting unconscious integration of numeric information is anecdotal. Regardless, we believe that the method used in unconscious arithmetic is useful for the study of the unconscious integration of two subliminal numbers. Recently, the same-different task has been regarded as a perfect tool for studying unconscious information integration [15]. It provides a chance to study the unconscious integration of numbers with a new method. Using this task, some studies found that the two unconscious stimuli presented simultaneously (such as two arrows pointing in the same/different directions, two same/different letters, two words referring to things belonging to the same/different categories, and two faces of same/different emotional valences) can be processed and integrated into a new representation [16,17,18,19]. For example, Tu et al. [17] subliminally presented two words of the same or different categories as the prime, followed by two words of the same or different categories as the target. Participants were asked to judge whether the two target words belonged to the same or different categories. The results revealed that the category relation of the two prime words could affect the judgment on the category relation of the things in the target. In some unconscious priming experiments using a single number as the prime, participants classified the number as larger or smaller than “5”. This is a semantic classification task that was similar to identifying an arrow as pointing to the left or right or classifying a word as a fruit or tool word in the same-different task. Therefore, the same-different task can be adapted to studying the unconscious integration of two subliminal numbers.

In a typical subliminal same-different task, two masked side-by-side primes are followed by two side-by-side targets [15]. Specifically, in this study, a pair of masked numbers serving as the prime was presented and then followed by another pair of numbers as the target. Participants were asked to decide whether the two target numbers were on the same or different sides of 5 in magnitude. In this task, there are three possible sources of priming effects: visual feature priming, categorical priming, and response priming [17]. Specifically, the visual feature priming effect was assessed by comparing a prime-to-target visual feature incongruent condition (e.g., “1–3” → “2–4”, the first pair of numbers was the prime, and the second pair was the target) with a visual feature congruent condition (e.g., “2–4” → “2–4”) while keeping the category and response constant across the prime and target in these two conditions. The category priming effect was assessed by comparing a prime-to-target category incongruent condition (e.g., “1–3” → “6–8”, with the two prime numbers on the smaller side of 5, and the two target numbers on the larger side of 5) with a category congruent condition (e.g., “7–9” → “6–8” with both pairs of numbers on the larger side of 5) while keeping the visual features in both cases changed and response constant across the prime and target in these two conditions. The response priming effect was evaluated by comparing a prime-to-target response in the incongruent condition (e.g., “1–6” → “7–9” with the two prime numbers on different sides of 5, but the target numbers on the same side of 5) with response in the congruent condition (e.g., “2–4” → “7–9” with both pairs on the same side of 5, albeit one being on the smaller and the other on the larger side) while keeping the visual features and category changed across the prime and target in these two conditions.

It should be noted that only the response priming effect can reflect the unconscious integration of the categorical relation of the two unconscious prime numbers because the participants’ response was about whether the two target numbers were on the same (in the same category) or different sides of 5 (in different categories) in magnitude. However, the three possible sources of priming effects were all analyzed in the present study for two reasons: (1) The change in visual features and categories across the prime and target must be controlled to maintain comparability between different conditions in response priming, necessitating the introduction of visual feature and category priming manipulations. (2) Our experimental design allowed for exploring the combined impact on the behavioral priming effects from multiple sources simultaneously. This is also addressed in the Discussion Section.

If the processing of unconscious integration of numbers occurred, it would be reflected in the response priming effect (Experiment 1). By the way, Experiment 2 was designed to eliminate the potential confounding influence of a category priming effect caused by one- versus two-sided change in the number category that might lead to a significant response priming effect in Experiment 1.

### 1.2. NCE in the Context of Unconscious Integration

The prime-to-target interstimulus interval (ISI) was manipulated as a factor to investigate the possible mechanism of attention in the unconscious processing of the categorical relation of two numbers using the same-different task. Previous research found that under the short prime-to-target ISI, participants responded consistently more quickly and more accurately when the prime and target were compatible than when they were incompatible (positive compatibility effect, PCE) [20]. In contrast, when the prime-to-target ISI was relatively long, the reaction time was longer or the accuracy was lower when the prime and target were compatible than when they were incompatible (negative compatibility effect, NCE) [21,22,23,24]. NCE was traditionally explained by self-inhibition, object-updating, and mask-triggered inhibition hypotheses [25]. However, when studying unconscious integration, it was observed in several studies that the NCE occurred no matter whether the prime-to-target ISI was short or long [16,17], which required a new explanation.

By examining the stimulus type across different studies of unconscious integration, we found that the unusual NCE occurred under relatively short prime-to-target ISI (PCE should appear according to the traditional view) when stimulation required complex, higher-level processing (e.g., faces in Liu et al.’s study [16] and words in Tu et al.’s study [17]). However, when stimulation was relatively simple and only required low-level processing (e.g., arrows in Wang et al.’s study [19]), the PCE, not NCE, was observed when the prime-to-target ISI was relatively short, which was consistent with the traditional view. We suspect that the unusual NCE under relatively short prime-to-target ISIs may be related to the availability of attentional resources [16,17]. Specifically, unconscious processing of a categorical relation between two relatively abstract, complex primes may need to engage more attentional resources compared with processing of two relatively simple primes or a single-stimulus prime. If the categorical relation in the target is incongruent with that in the prime, it might be easier to escape from the attention capture evoked by the prime compared with the condition in which the categorical relation in the prime and target is congruent. The ISI factor included in this study provided an opportunity to investigate the possible attentional mechanism underlying the unconscious integration of two numbers. It is speculated that processing of numbers, as a special kind of text content, may not be as intuitively simple as processing arrow symbols but may be similar to the unconscious integration of words as prime stimuli. Therefore, we suspect that the NCE will occur no matter whether the prime-to-target ISI is short or long.

Overall, this study tried to investigate unconscious integration of numbers with a new method (i.e., same-different task) and discussed the possibly abnormal NCE (i.e., NCE might occur under relatively short prime-to-target ISI) in the context of unconscious integration of information. In this study, we can see whether the categorical relationship between the two unconscious prime numbers (which is based on the unconscious integration of the two numbers) will affect the judgment on the categorical relationship between two target numbers. The results will potentially provide evidence for the unconscious integration of two subliminal numbers, which is fundamentally different from processing a single number. It is expected that this research can provide further evidence for the processing of numbers in particular as well as for unconscious integration theory in general.

## 2. Experiment 1

This experiment investigated whether the subliminal size relation of the two prime numbers (same side: both numbers were larger or smaller than 5; different sides: one number was larger and the other smaller than 5) could influence the responses regarding the supraliminal size relation of the two target numbers (on the same or different sides of 5).

### 2.1. Method

#### Participants

Twenty-five participants (mean age = 20.9 years, SD = 2.1 years, fifteen males) from the Guizhou University of Finance and Economics volunteered for this experiment. All participants were right-handed, had no history of neurological or psychiatric conditions, and had normal or corrected-to-normal vision. Informed consent was obtained before the experiment and all were paid for their participation. This study was approved by the IRB of Guizhou University of Finance and Economics.

Using G*Power software (v3.1), the post hoc power (1−β) approached 1.00 with the effect size *f* determined by the lowest partial η^2^ of about 0.2 when the effect was significant in the ANOVAs, the final number of participants (21) included in the analysis, and the alpha value 0.05.

### 2.2. Materials and Design

Eight numbers were used as the experimental stimuli (1 to 4, 6 to 9). During the experiment, two paired numbers were displayed side by side centrally on a uniform gray background, and each number subtended approximately 1.7 (height) × 1.2 (width) degrees of visual angle. The interval between the two numbers was about 3.0 degrees of visual angle.

Previous studies have shown that the proximity of distance in digital processing can affect the response time to stimuli with a larger distance between the prime and the target producing a smaller priming effect (e.g., pronouncing 7 is faster when it is after a prime of 6 than a prime of 5), an effect known as the distance priming effect [26]. Therefore, we kept the two simultaneously presented paired numbers nonconsecutive but with the distance between them being constant. Four combinations, i.e., “1–3”, “2–4”, “6–8”, and “7–9”, were selected for the same-side size relation condition. Similarly, “1–6”, “2–7”, “3–8”, and “4–9”, were selected for the different-sides size relation condition. Additionally, the left–right display sides of the two paired numbers were balanced throughout the experiment.

There were four conditions regarding the matching of the magnitude relation between the prime and the target: S-S, D-D, D-S, and S-D (see Table 1). The left letter stood for the magnitude relation of the two prime numbers and the right letter stood for the magnitude relation of the two target numbers. For example, in the S-D condition, the left letter S meant that the two numbers in the prime were on the same side of 5, e.g., “1 3” or “7 9”. The right letter D indicated that the two numbers in the target were on different sides of 5, e.g., “1 6” or “4 9”. In Table 1, under each of the prime-to-target magnitude relation transition types on the row headed by “Number pairing”, the two pairs of subscripted numbers in each column denoted the size of each number relative to 5 in the prime represented by the left pair of numbers and in the target represented by the right pair of numbers. In these pairings, G indicates that the digit is greater than 5 and L represents that the digit is less than 5. For each pair of digits in the prime or the target (e.g., G_6_L_1_), the left letter G indicated that a digit greater than 5 was on the left side of the fixation point and the right letter L indicated that a digit less than 5 was on the right side of the fixation point. The subscript numbers of the letters G and L indicated specific example numbers.

There were three subtypes in S-S and D-D (see Table 1), respectively, which were used to illustrate the visual feature, category, and response manipulation. The three lowercase letters in the parentheses represented altered (a) or unaltered (u) visual features, categories, and responses from the prime to the target. In the first subtype of S-S [S-S(uuu) in Table 1], two numbers were the same across the prime and target, so the visual features, category, and response were all unaltered across the prime and the target (e.g., “L_1_L_3_ → L_1_L_3_”). In the second subtype of S-S [S-S(auu) in Table 1], the numbers in the prime and the target were different but the categories of the numbers were the same, which led to visual features altered across the prime and target, but the category and the response held unaltered across the prime and the target (e.g., “L_1_L_3_ → L_2_L_4_”). In the third subtype of S-S [S-S(aau) in Table 1], the visual features and categories of the two paired numbers were both altered across the prime and target, but the response was unaltered (the response was “same side” both for the prime and the target) (e.g., “L_1_L_3_ → G_7_G_9_” or “G_6_G_8_ → L_2_L_4_”). Likewise, there were three subtypes of D-D [i.e., D-D(uuu), D-D(auu), and D-D(aau), see Table 1].

To obtain at least 64 trials for each condition in the visual feature, category, and response priming effect analyses, each pairing received different numbers of trials, which are shown in the parentheses next to the code letters (see Table 1).

To better understand the results in this design, it is necessary to first know that only the response priming effect can reflect the unconscious integration of the category relation of the two unconscious prime numbers. The other two priming effects, i.e., the visual feature and the category priming effects, can be regarded as control factors in the analysis for the response priming effect. In order to obtain the response priming effect, we compared the responses between altered and unaltered response conditions while controlling the changes of visual feature and category across prime and target (putting them at a consistent level of altered or unaltered states, see Table 1). A similar logic was used in the analyses of the visual and category effects.

### 2.3. Procedure

The sequence of events in each trial is displayed in Figure 1. First, a fixation cross appeared in the center of the screen for 300–600 ms with the exact time randomly determined. Subsequently, a pair of numbers serving as primes were presented side by side on the screen for 32 ms, followed by a backward mask for 50 ms or 150 ms. Then, the target stimuli were displayed on the screen until the participants responded or for 2000 ms, whichever happened first. The participants were asked to decide as fast and as accurately as possible whether the two target numbers were on the same or different sides of 5 by pressing key “1” or “2” with their right index and middle fingers, respectively. The mapping of the keys to the “same” and “different” responses was counterbalanced across participants. Next, participants reported their subjective experience of the prime visibility on the 4-point perceptual awareness scale (PAS): (1) ‘‘No experience’’, (2) ‘‘Brief glimpse’’ (a feeling that something appeared but nothing more specific than that), (3) ‘‘Almost clear experience’’, (4) ‘‘Absolutely clear experience’’ [27]. Finally, a blank screen appeared for 1000 ms. There were two blocks for each prime-to-target ISI condition. Each block had 256 trials and the different conditions in each block were displayed in a random order. The blocks from different ISI conditions were completed separately. Between the blocks, the participants could have a short rest.

Finally, participants were asked to complete a forced-choice prime visibility task (for the relationship between PAS and forced-choice task, see [28,29]). The procedure of this task was similar to that in the main priming experiment (i.e., with the numbers masked) except that the two masked numbers were the same, and the target was replaced by two options: the masked numbers were larger than 5 or smaller than 5. Participants were asked to determine/guess the answer between two options. The two choices (i.e., “smaller” and “larger”) remained on the screen until the participants made a response. There were 64 trials for this task. Before performing this task, participants were informed that only accuracy, and not speed, of the response was important. The masked number stimuli and the task were different from the main experiment, thus preventing a direct comparison between the state of awareness of masked numbers in the main experiment and the forced-choice task. The results of the forced-choice visibility test are still relevant for demonstrating null awareness of the masked prime stimuli. If the participants could not recognize the masked numbers in the visibility test condition, then we can infer that they should also be unable to recognize the size relationship between two masked numbers in the main experiment.

### 2.4. Results

#### 2.4.1. Prime Visibility Results

Four of the twenty-five participants were excluded from the data analysis because their individual mean percentage of correct recognition was higher than 60% (i.e., above chance level, by a binomial test, *p* < 0.05) in the forced-choice task. However, the analysis based on all data showed the results consistent with the one with the four participants excluded. Of the remaining twenty-one participants, no one selected ‘‘Almost clear experience’’ and ‘‘absolutely clear experience’’ of the PAS more than four times in each ISI condition. Under the conditions of 50 ms and 150 ms ISIs, the proportions of times of pressing “No experience” were 98.84% and 84.53%, respectively, while the proportions of pressing “Brief glimpse” were 1.12% and 15.41%, respectively. It appeared that as the presentation duration of the masking stimuli increased, participants might have perceived visual features of the masking stimuli as some form of meaningful stimuli.

Because the degree of unawareness of the masked number can raise serious questions about the unconscious priming [30,31], the Bayesian test, which is the most appropriate for assessing evidence in favor of a null hypothesis [32], is used here to test the chance level of the forced-choice prime visibility task (also the null effect in Experiment 2). The mean percentages of correct recognition were 49.14% under the 50 ms ISI condition and 48.24% under the 150 ms ISI condition in the forced-choice prime visibility test. In the Bayesian test by JASP software (v0.18.3.0), H1 was modeled with a normal distribution with mean (0.5) and standard deviation (SD), which was set to 0.05 based in previous studies using a forced-choice task. The Bayes factors BF_01_ were 49.838 and 205.350 under the conditions of 50 ms and 150 ms ISI, respectively.

#### 2.4.2. The Visual Feature Priming Effect

The visual feature priming effect was assessed by comparing the mean RT and accuracy of S-S(auu) with that of S-S(uuu) and, also, by comparing D-D(auu) with D-D(uuu) (see Table 1). In each case, the response was compared between altered and unaltered visual feature priming, with the other two factors, i.e., category and response, kept constant between the two compared conditions.

The mean RTs for correct responses were submitted to a 2 (visual feature change from prime to target: altered versus unaltered) by 2 (size relation between two target numbers: same versus different sides of 5) by 2 (ISI: 50 ms and 150 ms) three-way repeated-measures ANOVA (see Figure 2). The results showed that the main effect of the visual feature change was not significant, *F*(1, 20) = 0.342, *p* = 0.565, *η_p_*^2^ = 0.017. The main effect of size relation between two target numbers was significant in that participants responded faster in the same-side target size relation condition (mean RT = 658 ms) than in the different-sides target size relation condition (mean RT = 688 ms), *F*(1, 20) = 8.382, *p* = 0.009, *η_p_*^2^ = 0.295. The main effect of ISI was not significant, *F*(1, 20) = 3.969, *p* = 0.060, *η_p_*^2^ = 0.166. The interaction between ISI and visual feature change was significant, *F*(1, 20) = 5.335, *p* = 0.032, *η_p_*^2^ = 0.211. The simple effect showed that when the ISI was 50 ms, the effect of the visual feature change was significant. Participants responded faster to the unaltered visual feature condition than to the altered visual feature condition, *F*(1, 20) = 7.48, *p* = 0.013. But when the ISI was 150 ms, the effect of the visual feature change was not significant, *F*(1, 20) = 1.39, *p* = 0.252. The other interactions were nonsignificant, *Fs* < 1.058, *ps* > 0.316.

A similar ANOVA on accuracy indicated that both the main effect of visual feature change [*F*(1, 20) = 0.004, *p* = 0.951, *η_p_*^2^ = 0.000] and ISI [*F*(1, 20) = 3.727, *p* = 0.068, *η_p_*^2^ = 0.157] were nonsignificant. However, the main effect of target size relation was significant with the accuracy in the different-sides target size relation condition higher than that in the same-side target size relation condition (see Table 1), *F*(1, 20) = 6.740, *p* = 0.017, *η_p_*^2^ = 0.252.

Briefly, there was a PCE of visual feature priming when the ISI was 50 ms. However, it could be seen from Figure 2 that the visual feature priming effect tended to be negatively compatible when the ISI was long (150 ms). At any rate, however, the visual feature priming effect could not reflect the unconscious integration of the category relation of the two unconscious prime numbers.

#### 2.4.3. The Category Priming Effect

The category priming effect was assessed by comparing the RT and accuracy of S-S(auu) with those of S-S(aau) and, also, comparing D-D(auu) with those of D-D(aau) (see Table 1). In each case, the response was compared between altered and unaltered category priming, with the other two factors, i.e., visual features and response, kept constant between the two compared conditions.

The mean RTs for correct responses to the target were submitted to a 2 (category altered versus unaltered across prime and target) by 2 (size relation between two numbers in target: same versus different sides of 5) by 2 (ISI: 50 ms and 150 ms) three-way repeated-measures ANOVA (see Figure 3). The results showed that the main effect of the category change was significant, *F*(1, 20) = 10.292, *p* = 0.004, *η_p_*^2^ = 0.340. Participants responded faster in the prime-to-target unaltered category condition (mean RT = 675 ms) than in the altered condition (mean RT = 688 ms). The main effect of target size relation was significant, *F*(1, 20) = 4.715, *p* = 0.042, *η_p_*^2^ = 0.191. Participants responded faster in the same-side target size relation condition (mean RT = 669 ms) than in the different-sides target size relation condition (mean RT = 693 ms). Neither the main effect of ISI nor all the interactions were significant, *Fs* < 1.803, *ps* > 0.194. These results indicate a PCE for the category.

A similar ANOVA on accuracy indicated that there were no significant effects except for a significant main effect of target size relation with the accuracy in the different-sides target size relation condition higher than that in the same-side target size relation condition (see Table 1), *F*(1, 20) = 7.012, *p* = 0.015, *η_p_*^2^ = 0.260.

#### 2.4.4. The Response Priming Effect

The response priming effect was assessed by comparing the mean RT and accuracy of S-S(aau) with the RT of D-S(aaa) and, also, by comparing D-D(aau) with S-D(aaa) (see Table 1). In each case, the response was compared between altered and unaltered responses with the other two factors, i.e., the visual features and category, kept constant between the two compared conditions.

The mean RTs for correct responses to the target were submitted to a three-way repeated-measures ANOVA, with response change from prime to target (altered versus unaltered), size relation between two target numbers (same versus different sides of 5), and ISI (50 ms and 150 ms) as factors (see Figure 4). The main effect of the response change was significant, with the RT to the altered response condition (mean RT = 675 ms) being faster than the unaltered response condition (mean RT = 688 ms), *F*(1, 20) = 12.246, *p* = 0.002, *η_p_*^2^ = 0.380). The main effect of the target size relation was marginally significant, with the response in the same-side target size relation condition (mean RT = 669 ms) faster than in the different-sides target size relation condition (mean RT = 691 ms), *F*(1, 20) = 3.366, *p* = 0.081, *η_p_*^2^ = 0.144. None of the other main effects and interactions was significant, *Fs* < 2.472, *ps* > 0.132). In a word, an NCE was observed for the response factor.

A similar ANOVA on accuracy revealed no significant effect except for a significant main effect of the target size relation with the accuracy in the different-sides target size relation condition higher than that in the same-side target size relation condition (see Table 1), *F*(1,20) = 10.663, *p* = 0.004, *η_p_*^2^ = 0.348.

## 3. Experiment 2

In Experiment 1, a response NCE was observed, in which participants responded faster in the altered response condition than in the unaltered response condition. However, in the analysis for the response priming effect, the category of one number changed across the prime and target on one side in the D-S(aaa) and S-D(aaa) conditions (e.g., L_1_G_6_ → G_7_G_9_, left number 1 smaller than 5 → 7 larger than 5; L_1_L_3_ → L_2_G_7_, right number 3 → 7), but the categories of two numbers changed on both sides in the S-S(aau) and D-D(aau) conditions (e.g., S-S: L_2_L_4_ → G_6_G_8_; D-D: L_1_G_6_ → G_7_L_2_). Participants responded faster to D-S(aaa) and S-D(aaa) (category changed on one side only) than to S-S(aau) and D-D(aau) (categories changed on both sides). Therefore, the observed response NCE might have been confounded by a potential PCE due to one- versus two-sided change in the number category. Indeed, a PCE of category priming was observed in Experiment 1. However, this category PCE in Experiment 1 was based on a comparison of two-sided number category change with no change in the number category. Therefore, it was also possible that a comparison of one- versus two-sided change in the number category would not result in a significant category priming effect and would not confound the response NCE in Experiment 1. This is an empirical question, and hopefully, Experiment 2 will provide the answer.

In Experiment 2, the possibility of the category priming effect caused by one- versus two-sided change in the number category was investigated. If the response NCE in Experiment 1 was a spurious result from a category PCE, then, the response would be faster in the one-sided category change condition than in the two-sided category change condition in Experiment 2. Conversely, if no significant category PCE priming is observed in Experiment 2, we can have confidence in the response NCE obtained in Experiment 1.

### 3.1. Method

#### Participants

Twenty participants (mean age = 20.5 years, SD = 2.4 years, twelve males) from the Guizhou University of Finance and Economics volunteered for this experiment. None of them took part in Experiment 1. All participants were right-handed, had no history of current or past neurological or psychiatric conditions, and had normal or corrected-to-normal vision. They gave their informed consent before the experiment and were paid for participation. This study was approved by the IRB of the Guizhou University of Finance and Economics.

The results of Experiment 1 revealed that this study did not require a large number of participants in order to achieve enough power, so the number of participants was reduced slightly in Experiment 2. The post hoc calculation also showed that the number of participants in Experiment 2 met the requirements of the desired power. The post hoc power (1-β) approached 1.00 with the effect size determined by the partial *η*^2^ of about 0.5 for a significant effect, the final number of 18 participants included in the analysis, and the alpha value 0.05.

### 3.2. Materials and Design

The materials in Experiment 2 were the same as in Experiment 1 except that mosaic pictures, such as 
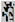
, were used in the one-sided number category change condition (S1 and D1 conditions in Table 2). The participants could not recognize the mosaic pictures. The mosaic pictures were used to keep the visual features altered on both sides across the prime and the target while al-lowing control over the number of categories changed across the prime and target.

Regarding the size relation of the two numbers in the target and the number of changed sides in the category across the prime and target, there were four conditions (see Table 2): S1, D1, S2, and D2. The left letter of each of the four notations stood for the size relation of the two numbers in the target, and the right number indicated the number of changed categories across the prime and the target. For example, in “S1”, the “S” denoted that both of the two numbers in the target were larger or smaller than 5, e.g., “2 4” or “7 9”; the “1” meant that only one side changed in the category across the prime and target, e.g., “1 ×”→“6 8” indicated that the category of the left number in the prime changed from smaller than 5 (“1”) to larger than 5 (“6”) in the target. Similarly, in “D2” (see Table 2), the “D” denoted that in the target, one number was larger than 5 and the other smaller than 5 (different sides of 5, e.g., “1 6” or “3 8”), and the “2” denoted that the category of the number changed on both sides across the prime and target, e.g., “7 2” → “3 8”. For each pair of numbers in the prime or the target in Table 2, G represented the number greater than 5, and L the number less than 5, which were the same as in Experiment 1. The symbol “×” stood for the mosaic.

The effect of the number of changed categories can be assessed by comparing responses to S1 with responses to S2 and comparing responses to D1 with responses to D2 (see Table 2). For the statistical analysis, a 2 (size relation between the two target numbers: same versus different sides of 5) by 2 (number of sides of changed categories between prime and target: one side versus two sides) by 2 (ISI: 50 ms and 150 ms) three-way repeated-measures ANOVA was conducted.

### 3.3. Procedure

The experimental procedure was the same as in Experiment 1. There were two blocks for each ISI condition. Each block had 128 trials with 32 trials for each condition displayed in a random order. After the participants finished the above experiment, they performed a forced-choice discrimination task, which was the same as in Experiment 1.

### 3.4. Results

#### 3.4.1. Prime Visibility Results

Two of the twenty participants were excluded from the data analysis because their individual mean percentage of correct recognition was higher than 58% (i.e., above chance level, by a binomial test, *p* < 0.05) in the forced-choice visibility task. However, the analysis that included the data from these two participants showed results consistent with the one without them. In addition, of the remaining eighteen participants, no one selected ‘‘almost clear experience’’ or ‘‘absolutely clear experience’’ on the PAS more than three times for each ISI condition. Under the conditions of 53 ms and 163 ms ISIs, the proportions of times of pressing “No experience” were 99.39% and 79.88%, respectively, while the proportions of pressing “Brief glimpse” were 0.52% and 19.92%, respectively. It also indicated that participants might have perceived visual features of the masking stimuli as some form of meaningful stimuli.

The mean percentages of correct recognition were 49.56% under the 50 ms ISI condition and 49.17% under the 150 ms ISI condition in the forced-choice prime visibility test. In the Bayesian test by JASP software, H1 was modeled with a normal distribution with mean (0.5) and standard deviation (0.05). The Bayes factors BF_01_ were 17.293 and 25.599 under the conditions of 50 ms and 150 ms ISI, respectively.

#### 3.4.2. The Effect of Number of Changed Categories

The RT and accuracy for each condition are displayed in Table 2. The mean RTs for the correct responses to the target were submitted to a three-way repeated-measures ANOVA with size relation between two target numbers (same versus different sides of 5), the number of changed categories across the prime and target (one side versus two sides), and ISI (50 ms, 150 ms) as within-subjects factors (see Figure 5). The main effect of target size relation was significant, *F*(1, 17) = 16.641, *p* = 0.001, *η_p_*^2^ = 0.495. Participants responded faster in same-side conditions (mean RT = 823 ms) than in different-sides conditions (mean RT = 910 ms). The main effect of the number of changed categories was not significant, *F*(1, 17) = 0.469, *p* =.502, *η_p_*^2^ = 0.027. No other effects were significant, *Fs* < 2.032, *ps* > 0.172. In addition, after we divided the data into two halves, the first and the second half, the results were consistent with the above results and did not show order effects.

In addition, using JASP software (v0.18.3.0), a three-way repeated-measures Bayesian ANOVA with default uniform priors was used to assess the null effect of the number of changed categories. For the target factor, BF_10_ = 39.31 supported very strong evidence for H1; for the ISI factor, BF_10_ = 0.732, showed anecdotal evidence for H1; for the factor of the number of changed categories, BF_10_ = 0.330 also indicated anecdotal evidence for H1. Thus, the results supported the null effect of the number of changed categories.

A similar ANOVA on accuracy revealed no significant effects, all *Fs* < 2.006, *ps* > 0.175. In short, there was no significant difference between the one-sided and two-sided category change conditions. Therefore, the response NCE obtained in Experiment 1 probably was not due to a category priming effect.

## 4. Discussion

The present research revealed that the digits could be categorized unconsciously as larger or smaller than 5 (as shown by the significant category priming effect in Experiment 1). Most importantly, after eliminating the potential confounding influence of a category priming effect arising from one- versus two-sided change in the number category in Experiment 2, the significant response priming effect in Experiment 1 indicated that the category relationship of the two unconscious prime numbers could be processed and could affect the judgment about the category relationship of the two target numbers. These results not only reflect the unconscious extraction of digital semantics itself but also support the idea of unconscious integration of two subliminal numbers. In addition, the fact that the response NCE was independent of the length of ISI suggested a possible role of attention in the unconscious information integration in the same-different task paradigm.

### 4.1. Unconscious Integration of Numbers

Exploring the scope and complexity of unconscious processing is an important goal in the field of unconsciousness research. The findings of the unconscious integration of numbers in the present study refuted the early belief that only simple numbers can be processed unconsciously and supported the possibility of the still-debated complex, unconscious processing of numbers. Several studies indicated that individuals could perform simple unconscious arithmetic such as addition [8] and single-digit multiplication [9], which entails unconscious information integration. Furthermore, Sklar et al. [33] argued that complex single-digit subtraction could also be processed unconsciously. For example, they found that participants responded significantly more quickly when the prime equation was congruent with the target (prime 9-3-4, “-“ being minus, and target 2) than incongruent with the target (prime 9-3-4, and target 5). However, follow-up research [13] and a reexamination of the data from Sklar et al.’s studies [14] suggested that evidence supporting unconscious arithmetic was weak. The results of the present study confirmed the possibility of unconscious processing of numbers at a deeper level than single digits.

Importantly, the unconscious integration of the category relationship of the two subliminal prime numbers was reflected in the significant response priming effect. In the paradigm of this study, participants were asked to judge the size relationship of the two simultaneously presented target numbers (on the same or different sides of 5) instead of directly categorizing the number as greater or smaller than 5. The response to the target was affected by the magnitude relationship of the two subliminal prime numbers. It was consistent with the results of previous studies that found that people could process the relationship of two simultaneously presented unconscious stimuli, including the pointing direction congruence between two masked arrows [19,34], the sameness of the shapes of two objects [35] or two letters [18], category congruence between two masked words [17], and emotional valence congruence between two masked faces [16]. Overall, it seems that unconscious integration is a general brain processing mechanism for a wide range of stimulus types.

### 4.2. Attentional Modulation of the NCE within the Context of Unconscious Integration

Although there was a significant response priming effect, it was an NCE no matter whether the prime-to-target ISI was short or long. This contradicts the conventional view that PCE should be observed when the prime-to-target ISI is relatively short [20], whereas NCE should occur when the prime-to-target ISI is relatively long [21,24]. In the study of unconscious information integration, the findings concerning this issue are inconsistent. To sum up, for the simultaneously presented low-level unconscious stimuli such as arrows [19], the shapes of objects [35], or letters [18], the extant findings are consistent with the traditional view, i.e., PCE occurs when the prime-to-target ISI is relatively short. However, when the unconscious stimuli are of a higher order, such as words [17] and emotional faces [16], the findings contradict the traditional view, i.e., NCE occurs when the prime-to-target ISI is relatively short. In this study, the numbers were processed in a more complex way, similar to processing words. The participants were asked to judge the category relationship between two numbers, which was likely to involve a higher level of semantic processing. The results again showed that an NCE occurred when the prime-to-target ISI was relatively short.

The NCE is traditionally explained by three different hypotheses: self-inhibition, object-updating, and mask-triggered inhibition hypotheses [25]. However, we do not think they can explain the NCE in this study. For the self-inhibition hypothesis, one of the conditions for self-inhibition to occur is that the delay between the prime and target is long enough for this inhibition to become effective [22], which cannot explain the NCE of response priming under the short ISI condition (50 ms in the present study, and even 0 ms in Liu et al.’s study [16]). The object-updating hypothesis suggests that NCE is triggered by masking that contains features calling for the response opposite to the prime. In the present study, the masking stimulus was not derived from number transformation. In addition, the NCE of response priming independent of ISI was also observed in other two studies about unconscious integration [16,17] where masking was totally different from the prime and target. Therefore, the object-updating hypothesis fails to explain the results of this study. For another mask-triggered inhibition hypothesis, the inhibition is predicted to be time-locked to the mask, not the prime in the self-inhibition hypothesis [36]. It seems that this hypothesis also cannot explain the NCE of response priming under the condition of 0 ms ISI, in which there was no mask between the prime and target, as in Liu et al.’s study [16].

In the present study, attention resources may also be one of the causes of the NCE when the ISI is relatively short. There may be a difference in cognitive processing between the unconscious processing of the category of a single number and that of the categories of two numbers. Exploring the scope of complexity of information that can be processed unconsciously is an important step toward understanding the scope and mechanism of unconscious processing. Compared with the unconscious judging of the category of a single number, the unconscious integration of two numbers presented simultaneously may be more complex and require greater semantic engagement and attention resources [37]. When a higher degree of attentional resources is engaged in the processing of the category relation of the two prime numbers (in comparison with the processing of low-level stimuli), it can be easier to escape from the attention capture evoked by the processing of the category relation between the two prime numbers under the prime-to-target incongruent condition compared with the congruent prime-to-target condition [16,17].

### 4.3. Priming Effects from Multiple Sources

There was another interesting result worth discussing. Although the response priming effect was observed, which was the most relevant to our purpose, the visual feature priming and the category priming effects were also observed [17]. Specifically, the category priming effect in Experiment 1 indicated a processing sub-component (deciding on greater or smaller than 5) of the main task (deciding on the same or different sides of 5). The significant category priming effect confirmed the unconscious semantic processing of numbers [11,12]. More importantly, it also suggested a new unconscious processing mechanism—that is, as long as the unconscious processing sub-components were involved in a main task, which was necessary but secondary to the main task, the unconscious sub-processing might also have unconscious effects on behavior. This influence path can explain some phenomena. For example, in creative activities, the solution to a creative problem is often triggered by a stimulus with similar but also seemingly unrelated attributes. Specifically, some sub-processing components of the seemingly unrelated task can unconsciously establish a connection with the main creative task, just like Watt invented the steam engine after being inspired by the boiling water pushing the lid up.

### 4.4. Another Possible Explanation

Finally, we wanted to discuss another explanation that seemed plausible. The number pairing such as “1–3”, “2–4”, “6–8”, and “7–9” was selected in the same-side size relation condition and such as “1–6”, “2–7”, “3–8”, and “4–9” was selected in the different-sides size relation condition. Because the distance between the two target numbers was correlated with the size relationship, i.e., the short distance (“1–3”, “2–4”, “6–8”, and “7–9”) indicated same-side size relation and the long distance (“1–6”, “2–7”, “3–8”, and “4–9”) indicated different-sides size relation, participants could change the response strategy from judging the size relation between the two target numbers to estimating the distance between the two target numbers. Post-experiment inquiry showed that the use of this kind of strategy was rare. Even if participants used this strategy, calculating the distance between two numbers is a process of information integration.

## 5. Conclusions

In conclusion, the results revealed that the numerals could be categorized unconsciously, and the category relationship of the two masked unconscious numbers could be processed, which reflected the unconscious integration of the two subliminal numbers. These results have important implications for the study of the potential mechanisms of unconscious processing of numerals and for furthering the understanding of the scope of unconscious information processing. They can also help us understand the psychological mechanisms behind learning, decision-making, mental health issues, etc., guiding practices in related fields.

## Figures and Tables

**Figure 1 behavsci-14-00296-f001:**
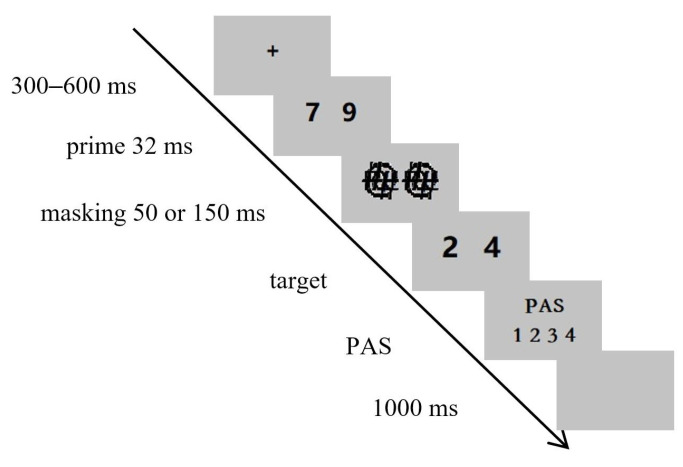
The sequence of events in Experiment 1.

**Figure 2 behavsci-14-00296-f002:**
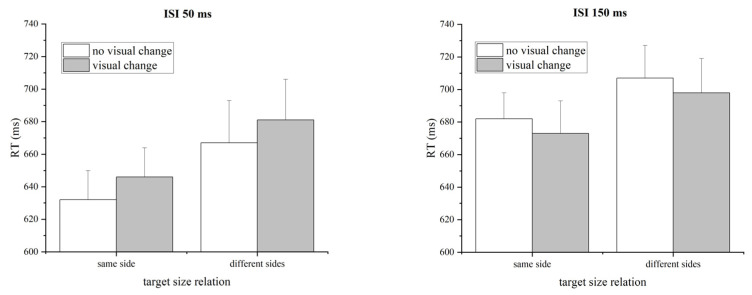
Mean RTs (ms) under each prime-to-target visual feature congruence condition for each target size relation and each ISI condition in Experiment 1. The results show a significant PCE of visual feature priming when the ISI is 53 ms. The figure also shows that the visual feature priming effect tends to be negatively compatible when the ISI is 150 ms. The error bars represent one standard error of the mean.

**Figure 3 behavsci-14-00296-f003:**
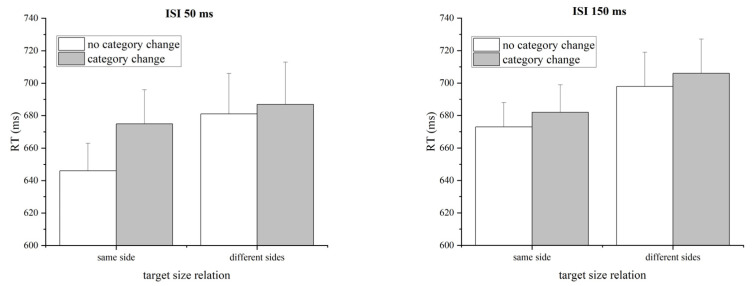
Mean RTs (ms) under each prime-to-target category congruence condition for each target size relation and each ISI condition in experiment 1. The results reveal a PCE for the category factor. Error bars represent the standard error of the mean.

**Figure 4 behavsci-14-00296-f004:**
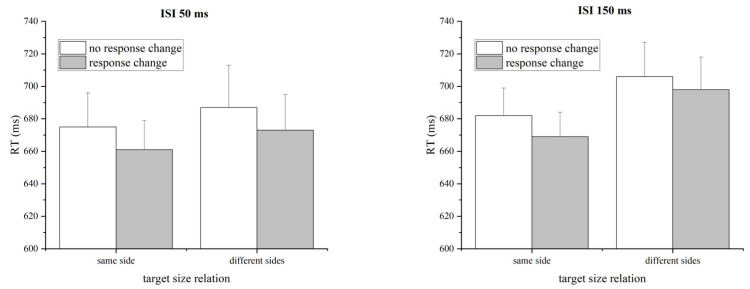
Mean RTs (ms) under each prime-to-target response congruence condition for each target size relation and each ISI condition in Experiment 1. The results show an NCE for the response factor. Error bars represent one standard error of the mean.

**Figure 5 behavsci-14-00296-f005:**
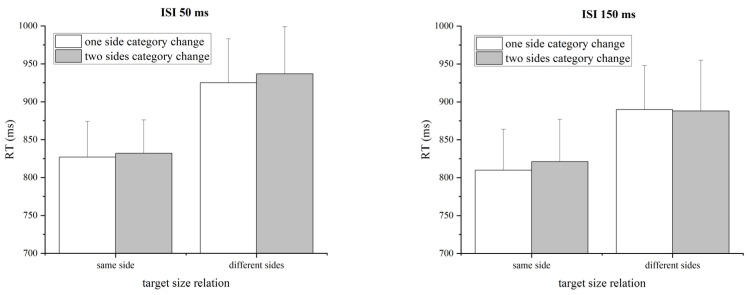
Mean RTs (ms) under each (one or two) prime-to-target category change condition for each target size relation and each ISI condition in Experiment 2. The results do not show a significant difference between the one-sided category change and the two-sided category change condition. Error bars represent the standard error of the mean.

**Table 1 behavsci-14-00296-t001:** The details of the pairing and results [mean RTs and accuracy (ACC) ± SE] in each condition for the analyses of visual feature, category, and response priming effects.

Conditions		S-S			D-D			D-S (aaa)	S-D (aaa)
		S-S (uuu)	S-S (auu)	S-S (aau)	D-D (uuu)	D-D (auu)	D-D (aau)		
Number pairing		L_1_L_3_ → L_1_L_3_ (32) G_8_G_6_ → G_8_G_6_ (32)	L_1_L_3_ → L_2_L_4_ (32) G_6_G_8_ → G_9_G_7_ (32)	L_1_L_3_ → G_7_G_9_ (32) G_8_G_6_ → L_2_L_4_ (32)	L_1_G_6_ → L_1_G_6_ (32) G_7_L_2_ → G_7_L_2_ (32)	L_2_G_7_ → L_3_G_8_ (32) G_6_L_1_ → G_7_L_2_ (32)	L_1_G_6_ → G_7_L_2_ (32) G_6_L_1_ → L_3_G_8_ (32)	L_1_G_6_ → L_4_L_2_ (16) L_2_G_7_ → G_6_G_8_ (16) G_8_L_3_ → L_2_L_4_ (16) G_7_L_2_ → G_6_G_8_ (16)	L_4_L_2_ → L_3_G_8_ (16) L_1_L_3_ → G_7_L_2_ (16) G_7_G_9_ → L_1_G_6_ (16) G_9_G_7_ → G_8_L_3_ (16)
50 ms ISI	RT	651 (26)	659 (22)	675 (20)	677 (29)	687 (27)	687 (26)	661 (17)	673 (21)
	ACC	0.95 (0.01)	0.95 (0.01)	0.95 (0.01)	0.97 (0.01)	0.97 (0.01)	0.97 (0.01)	0.95 (0.01)	0.96 (0.01)
150 ms ISI	RT	685 (15)	677 (15)	682 (17)	708 (20)	703 (22)	706 (21)	669 (14)	698 (19)
	ACC	0.93 (0.01)	0.93 (0.02)	0.94 (0.01)	0.95 (0.01)	0.95 (0.01)	0.96 (0.01)	0.94 (0.01)	0.96 (0.01)

Note. The two uppercase letters S and D indicate the magnitude relation of the two prime (and target) numbers, respectively (S stands for “same side of 5”, and D for “different sides of 5” with the first letter representing the relationship of the two numbers in the prime and the second letter representing the relationship in the target). The three lowercase letters in the parentheses next to the capital letters S and D represent the changes or no change from the prime to the target in visual and category features and response, respectively (“u” stands for “unaltered”, and “a” for “altered”). On the row with “Number pairing”, the two pairs of letters represent the categories of the four numbers used in a trial, with the first pair being the prime, and the second pair the target. G represents that the digit is greater than 5, and L represents that the digit is less than 5. The left and right positions of the two letters in each pair represent the display position on the screen. And the subscript numbers represent specific example numbers. The two-digit number in the parentheses is the number of trials for each pairing condition.

**Table 2 behavsci-14-00296-t002:** The details of number pairing and the results (RT and accuracy mean ± SE) for each condition.

Conditions		S1	D1	S2	D2
Number pairing		L_2_×→ G_7_G_9_ (16) ×L_3_→ G_6_G_8_ (16) G_7_×→ L_1_L_3_ (16) ×G_8_→ L_2_L_4_ (16)	L_1_×→G_6_L_1_ (16) ×G_6_→G_7_L_2_ (16) ×L_2_→L_3_G_8_ (16) G_7_×→L_1_G_6_ (16)	L_1_L_3_→G_6_G_8_ (32) G_6_G_8_→L_2_L_4_ (32)	L_1_G_6_→G_7_L_2_ (32) G_7_L_2_→L_3_G_8_ (32)
50 ms	RT (ms)	827 ± 47	925 ± 58	832 ± 44	937 ± 62
	accuracy	0.97 ± 0.01	0.97 ± 0.01	0.97 ± 0.01	0.98 ± 0.01
150 ms	RT (ms)	810 ± 54	890 ± 58	821 ± 56	888 ± 67
	accuracy	0.98 ± 0.00	0.97 ± 0.01	0.98 ± 0.01	0.98 ± 0.00

Note. On the row headed by “Conditions”, the capital letter of each of the four notations stands for the size relation of the two numbers in the target, and the number next to it indicates the number of changed categories across the prime and the target. On the row headed by “Number pairing”, the symbol “×” stands for the mosaic, and the two pairs of letters (including the symbol “×”) represent the categories of the numbers relative to 5, with the first pair being prime, and the second pair the target. G represents that the digit is greater than 5 and L represents that the digit is less than 5. The left and right positions of the two letters in each pair represent the display positions on the screen. The subscript numbers represent specific example numbers. The two-digit number in the parentheses is the number of trials for each pairing condition.

## Data Availability

The datasets generated and analyzed during the current study are available from the corresponding author upon reasonable request.

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
