# Peer review of "Unconscious Integration of Categorical Relationship of Two Subliminal Numbers in Comparison with “5”"

_behavsci, 2024, doi:10.3390/bs14040296_

Round 1

Reviewer 1 Report

Comments and Suggestions for Authors

Review of Ms. "Unconscious integration of categorical relationship of two subliminal numbers: On one or two sides of "5" in magnitude", by Changjun Li, Qingying Liu, Yingjuan Liu, Jerwen Jou, and Shen Tu

In two carefully designed experiments, the authors explore priming effects by numeral primes. The target consists of a pair of numbers, and the task is to make a same-different decision as to whether both numbers are on the same side of "5" or on different sides. Following this speeded decision, the visibility of the prime is judged by means of the "Perceptual Awareness Scale". The target is preceded by a number pair as a prime, and the authors follow an intricate design to make relevant comparisons regarding visual priming, category priming, and the crucial response priming. The major finding is a Negative Compatibility Effect when prime and target activate different motor responses. Experiment 2 explores a possible artifact in this tightly-woven design.

This is a very good paper, and I'm supportive of publication. However, I still have a few questions.

MAJOR POINTS

- Participants have to give a speeded same-different response and then perform the PAS rating on the same trial. By now many papers have shown that this dual-task situation does not only affect the rating, but also the speeded response (there are papers on that by Wentura, Rohr, Kiefer). Dual-task interference might explain the long response times: Participants have to wait out the target (especially with long SOAs) and then give a pre-planned sequence of same-different and PAS responses. That means they respond out of a working memory representation, not from the bottom-up flow of prime and target information, which can drastically alter the structure of the response priming effect and even reverse them (Biafora & Schmidt, 2022). In the current paper, the concurrent measurement of priming and PAS is not even necessary because different visibility ratings are not compared in the first place. -- In my opinion, the paper would greatly benefit from simple time-bin plots where effects are plotted against physical time of the response (not deciles, but timebins). Such plots would reveal whether the NCE is perhaps preceded a regular PCE (Panis & Schmidt, 2016) and whether the reversal in reaction times occurs early or late in the process. Maybe complex tasks are prone to show NCE effects simply because RTs are longer and are therefore more likely to be caught up in inhibition effects late in the trial (we concluded in a series of papers that the NCE is due to mask-induced inhibition of the primed response which occurs about 400 ms after mask onset, as originally proposed by Piotr Jaskowski).

- Exclusion of participants on the basis of prime visibility is a big methodological problem and should never be done. It creates regression to the mean and underestimates the visibility of the prime without actually changing it on the population level (Shanks, 2017; Schmidt, 2015). Thankfully, the number of excluded participants is relatively low here, but it would be better to keep them in the sample and accept the slightly elevated forced-choice performance. An alternative would be to make a formal comparison with those persons left in or out.

MINOR POINTS

The by-title line is strange.

Abstract: It does not become entirely clear where positive or negative priming effects are observed.

p3 §2: The list of references about the NCE is not adequate. The work by Boy and Sumner is absolutely relevant, but the effect was first discovered by Eimer & Schlaghecken and investigated in detail by people like Jaskowski, Verleger, Enns, and some others. Lingnau & Vorberg (2005) were the first to establish the full time-course of PCE and NCE. The theoretical discussion of the effect is similarly unspecific: the authors describe R&S's theory almost verbatim without referencing them.

p3 §2: It should be "in the absence of perceptual evidence for the PRIME", right? Again, that is the original Eimer and Schlaghecken theory, but there is a lot of evidence against it from variations of the mask (Jaskowski, Verleger) and examinations of the time-locking of the NCE to prime vs. mask (Panis & Schmidt).

p6 §4: The PAS doesn't really fit the task. The critical feature of the prime is the same-different relation, not the overall "visibility" of the "stimulus". From the point of view of our "Cue-Set Theory" of visibility measures, the relevant question would be whether participants can identify whether the two prime digits are on the same or on different sides of "5". Atas is really the wrong reference for the wording of the scale; obviously the original wording by Ramsoy and Overgaard is used here.

p10 final §: Don't understand: What are those "mosaic pictures" and what are they for?

p12 bottom: This is no adequate way to report Bayesian tests. Bayes factors are literally meaningless if we don't know the prior distributions that are compared under H0 and H1. Priors have to be suitably motivated, too. Keep in mind that a "Bayesian Anova" is no Anova at all: it is not based on variance decomposition, and each of the seven priors is designed independently. Therefore, each one must be described and motivated, and it should be transparent whether the priors are progressive or conservative with respect to H0. The hierarchical model into which they are placed must be described as well because there are many ways to implement this for repeated measures.

And too many sentences begin with "And".

Reviewer 2 Report

Comments and Suggestions for Authors

Overview: Hypotheses about unconscious integration of information were tested by comparisons of masked Arabic digit pairs in a priming paradigm with same-different judgments. Visibility of the masked stimuli was assessed by a perceptual awareness scale (PAS) on each trial and forced-choice lower/higher discrimination trials. The results showed a category priming effect and a response priming effect. The latter suggests an unconscious process that integrates the prime stimuli.

Positives: The topic of unconscious integration has great importance given that the primary challenge of explaining consciousness is how many small bits of information add up to form a holistic experience. The paper is well-written and nicely explains the scientific literature, including some controversies. The methods and statistics largely meet the standards expected for masked priming experiments.

Problem #1 - assessment of awareness PAS: The most challenging issue for any kind of masked priming experiment is to demonstrate that there was absolutely no awareness of the prime stimuli. The authors use the PAS (on each trial) and forced-choice discrimination (a special block of trials) of higher or lower than five for a single masked prime digit.

Unfortunately, there are significant problems. The PAS results were primarily used to eliminate a few participants who had high visibility. This is okay. The problem is that the actual PAS results from the 1 "no experience" and 2 "brief glimpse" are not reported. A common finding in masked priming is that people often choose the 2 rating, suggesting that they did not have total unawareness of the masked prime stimuli. When this occurs some slight conscious experience of the masked prime stimuli may have been sufficient to guide the responses.

Request: The authors should provide results for the PAS, especially for the 1 and 2 ratings. What was the average PAS rating? In accordance with the PAS literature, if 2 ratings are used often then it is unwarranted to claim that the participants were completely unaware of the stimuli. A PAS average closer to the two rating on this scale would suggest partial awareness.

Problem #2 - assessment of awareness forced-choice: The authors chose single digit stimuli (1 to 4 or 6 to 9) for this forced-choice awareness assessment, making the stimuli different from the experiments. In addition, the task was a smaller or larger than five judgment which is also different from the same-different task of the experiments. These differences make the experiment less relevant to the experimental conditions, which weakens the logic of a comparison between the awareness condition and the experiment. The authors state: "It is reasonable to assume that if participants could not recognize the two masked same numbers in this test, they should not be able to recognize the size relation between two different masked numbers." That's possible, but we really don't know. What would be much better is to have exactly the same conditions for the both the experimental trials and the awareness assessment trials so a direct comparison can be made with high certainty.

Request: The authors must acknowledge that using a different paradigm, both stimuli and task, for the forced-choice awareness assessment means that this condition cannot be directly compared to the experimental trials. Unfortunately, the results are only weakly relevant because this awareness condition is different from the main experiment. Delete the sentence about this comparison being "reasonable". The visibility testing results might be the same as the main experiment, but we cannot know this with certainty due to the different conditions. State that these results are merely suggestive and relevant, yet also uncertain.

Problem #3 - assessment of awareness, forced-choice statistical tests: The forced-choice awareness assessment is evaluated with non-significant t-tests. This was a standard procedure about 10 to 20 years ago when Dehaene and other researchers were working on this paradigm. The use of traditional (sometimes called frequentist) tests for awareness assessment has been significantly challenged in recent years. Reviews by Zoltan Dienes and Myron Tsikandilakis make a compelling argument that more appropriate approach is to use Bayesian tests instead of traditional tests. The basic rationale is that these experiments require evidence for a null hypothesis of no awareness and Bayesian tests are the most appropriate for assessing evidence in favor of a null hypothesis. Another relevant matter for this paper is that new studies using Bayesian methods on the Dehaene digit priming paradigm have raised serious questions about the validity of previous studies done with this method. The entire paradigm may have been based on an inappropriate use of traditional statistics for assessing masked prime awareness. These studies are a serious challenge to the methodology that the authors have based their work upon.

Meyen, S., Zerweck, I. A., Amado, C., von Luxburg, U., & Franz, V. H. (2022). Advancing research on unconscious priming: When can scientists claim an indirect task advantage? Journal of Experimental Psychology: General, 151(1), 65–81. https://doi.org/10.1037/xge0001065

Zerweck, I. A., Kao, C.-S., Meyen, S., Amado, C., von Eltz, M., Klimm, M., & Franz, V. H. (2021). Number processing outside awareness? Systematically testing sensitivities of direct and indirect measures of consciousness. Attention, Perception, & Psychophysics, 83, 2510–2529. https://doi.org/10.3758/s13414-021-02312-2

Request: Replace all of the visibility t-test results with Bayesian t-tests with BF01 for the null hypothesis. Choosing prior values for these Bayesian tests might be difficult. Dienes' papers have some suggestions about setting priors. The authors must also acknowledge the above papers by Meyen and Zerweck that raise serious questions about the degree of masked prime unawareness in digit priming paradigms.

Problem #4 - complicated design: Experiment 1 simultaneously tests for visual feature priming, categorical priming, and response priming. Although this ambitious goal is theoretically possible with a factorial design, this approach seems overly complicated. "It should be noted that only the response priming effect can reflect the unconscious integration of the categorical relation of the two unconscious prime numbers" If response priming is the key priming effect, then why was it necessary to examine visual feature priming and categorical priming too? The results were analyzed by three separate factorial ANOVA tests for priming effects. Some of the conditions (S-S(aau) and D-D(aau)) were used in more than one analysis (categorical and response priming). Repeated testing of conditions should generally be avoided in statistical testing because this can cause alpha inflation (cumulative p values going above .05).

Request: The rationale for simultaneously examining three forms of priming in a single experiment needs to be explained better. The appropriateness of multiple factorial ANOVA tests on one experiment needs to be evaluated to determine if this is problematic. Consider making one giant factorial ANOVA for the whole experiment because this the typical use of factorial ANOVA. A related problem to address for these comparisons is the unequal numbers of trials in the D-D (192), S-S (192), D-S (64), and S-D (64) conditions.

Problem #5 - interpretation of Experiment 2: Damian (2001) argued that consciously experienced subtle cues (non-semantic, possibly an integration of prime and mask) created a stimulus-response mapping that explained masked priming effects without needing an unconscious mechanism. In the present study, a "mosaic" stimulus was used in Experiment 2 to address a possible artifact of one-sided vs. two-sided stimuli in category priming obtained in Experiment 1. The authors interpret Experiment 2 as evidence that the negative compatibility effect in Experiment 1 is not problematic. However, an alternative view of this outcome is that no unconscious integration is needed. A somewhat extreme alternative interpretation is possible. The mosaic is non-numerical, so any priming that happens could suggest that numerical relationships between stimulus pairs are actually unimportant for this paradigm. Some other cues, perhaps non-semantic, are used to guide the task when these cues are used repeatedly over many trials. So, this could fit Damian's suggestion that subtle consciously experienced cues are really responsible for the priming effects. This viewpoint undermines the authors' interpretations. Maybe this critique based on Damian's view is far-fetched and unrealistic, but it should be addressed given the mosaic stimulus is non-numerical.

Request: The authors should consider that the priming effects in this paradigm are driving by consciously experienced subtle cues. Does Experiment 2 support this view, thereby contradicting the results from Experiment 1? The possibility of stimulus - response learning could be assessed by looking for order effects over the course of the experiment.

Problem #6 - insufficient documentation of Bayesian statistics: In Experiment 2: "In addition, the mean RTs were submitted to a three-way repeated-measures Bayesian ANOVA." This needs more rationale. Why were Bayesian statistics chosen here when all other tests in this paper are not Bayesian?

Request: An explanation of why Bayesian ANOVAs are used in experiment 2 should be provided. All Bayesian tests should include informed prior values or default priors and any other assumptions. These priors should be justified. This suggestion also applies to the recommendation of using Bayesian statistics for the awareness check conditions.

Round 2

Reviewer 1 Report

Comments and Suggestions for Authors

Review of Revised Ms. "Unconscious integration of categorical relationship of two subliminal numbers in comparison with '5' ", by Changjun Li, Qingying Liu, Yingjuan Liu, Jerwen Jou, and Shen Tu

The authors have addressed by comments convincingly and revised their manuscript accordingly. In particular, they have addressed my "major points" about dual tasks altering the temporal structure of the priming effect and about exclusion of participants. They also now discuss the theoretical positions regarding the NCE in more detail and better reference to existing theories. For the review process, they also provided survival plots of the RT distributions in some critical conditions, but I agree that there is nothing important to discover there. The minor points have been addressed as well, the most important of them concerning some references, the criterion content of the visibility test, and the specification of Bayes Factors. I am supportive of publication.

Author Response

There are no issues requiring a response. Thank you for the reviewer's affirmation.

Reviewer 2 Report

Comments and Suggestions for Authors

Problem #1 - assessment of awareness PAS: Reporting the percentages of responses for the #1 and #2 ratings on this scale is a big improvement that provides evidence for the null awareness of the prime stimuli. This addition satisfies my request for more PAS information.

Problem #2 - assessment of awareness forced-choice: Dropping the word “reasonable” from line 302 and adding that the results are suggestive was an improvement. However, the next sentences are still problematic. Line 305+: “However, because this paper focused on the integration of “unconscious information”, ensuring the unconsciousness of the masked numbers can be seen as a more primary criterion than the unconsciousness of the relationship between the two prime numbers.” This idea needs to be expressed more clearly. The authors are making a logical argument that if the simpler display in the awareness condition lacks awareness then the more complex situation in the main experiment should also be lacking in awareness.

Recommendations: “The same two masked numbers were different” Add that the task was different from the main experiment too. Delete the word “uncertain” from line 305 because that’s too negative. We did learn something useful from this forced-choice test that helps us to interpret the results from the main test.

Suggestion revision: “The masked number stimuli and the task were were different from the main experiment, thereby preventing a direct comparison between the awareness check and experimental conditions.

Delete the “However” sentence because it’s awkward and not very convincing.

Here is a possible revision based on the last sentence in the paragraph. The evidence here is an inference, not a direct demonstration, because a direct comparison is not possible.

Suggested revision: The results of the forced-choice visibility test are still relevant though for demonstrating null awareness of the masked prime stimuli. If the participants cannot recognize the masked numbers in the visibility test condition, then we can infer that they should also be unable to recognize the size relationship between two masked numbers in the main experiment.

Problem #3 - assessment of awareness, forced-choice statistical tests: The addition of Bayesian tests for null visibility plus the citation of the Zerweck and Meyen papers is a significant improvement that satisfies my request.

Problem #4 - complicated design: The authors have explained their reasons in the response to reviewer #2, but the actual paper still fails to really communicate clearly why address this issue of why feature priming, categorical priming, and response priming need to be pursued in one experiment. A small writing problem here is that the critical idea about response priming is located at the end of the paragraph. This writing problem is called “burying the lede.” This response priming idea needs to be more prominent because it is really the critical point of the experiment.

Suggested revision: On line 175, make “It should be noted that…” the start of a new paragraph. This is really important so it will receive more attention at the beginning of the paragraph rather than being at the end of the paragraph with many topics. This is a simple revision that will make this idea more clear.

Problem #5 - interpretation of Experiment 2: I am satisfied with this response.

Problem #6 - insufficient documentation of Bayesian statistics: The explanation and technical details of Bayesian statistics has been improved.
